# Urea Addition Promotes the Metabolism and Utilization of Nitrogen in Cucumber

**Chao Ma [1,2], Tiantian Ban [2], Hongjun Yu [3], Qiang Li [3], Xiaohui Li [2], Weijie Jiang [1,3,\*] and Jianming Xie [1,\*]**

1. College of Horticulture, Gansu Agricultural University, Anning District, Lanzhou 730070, Gansu, China; machao621@126.com
2. Institute of Horticulture, Guizhou Academy of Agricultural Science, Xiaohe District, Guiyang 550006, Guizhou, China; BanTianTianq123@163.com (T.B.); hvlerry@foxmail.com (X.L.)
3. Institute of Vegetables and Flowers, Chinese Academy of Agricultural Sciences, 12 Zhongguancun S. St., Beijing 100081, China; yuhongjun@caas.cn (H.Y.); liqiang05@caas.cn (Q.L.)
\* Correspondence: jiangweijie@caas.cn (W.J.); xiejianming@gsau.edu.cn (J.X.); Tel.: +86-10-82108797 (W.J.); +86-934-7632465 (J.X.)

**Abstract:** Nitrogen (N) forms include ammonium [$NH_4^+$-N], nitrate [$NO_3^-$-N], and urea [$CO(NH_2)_2$]. Urea is the most common nitrogen fertilizer in agriculture due to its inexpensive price and high N content. Although the reciprocal influence between $NO_3^-$-N and $NH_4^+$-N is well known, $CO(NH_2)_2$ interactions with these inorganic N forms have been poorly studied. We studied the effects of different nitrogen forms with equal nitrogen on dry matter, yield, enzyme activity, and gene expression levels in cucumber. $NO_3^-$-N treatment with equal $CO(NH_2)_2$ promoted nitrate reduction, urea utilization, and the GS/GOGAT cycle but reduced the nitrate content. *UR-2*, *NR-2*, *NR-3*, *NiR*, *GOGAT-1-1*, and *GS-4* were upregulated in response to these changes. $NH_4^+$-N treatment with equal $CO(NH_2)_2$ promoted nitrogen metabolism and relieved the ammonia toxicity of pure $NH_4^+$-N treatment. *UR-2*, *GOGAT-2-2*, and *GS-4* were upregulated, and *GDH-3* was downregulated in response to these changes. Treatment with both $NO_3^-$-N with added equal $CO(NH_2)_2$ and $NH_4^+$-N with added equal $CO(NH_2)_2$ enhanced the activities of GOGAT, GS, and UR and the amino acid pathway of urea metabolism; manifested higher glutamate, protein, chlorophyll, and nitrogen contents; and improved dry matter weight. A greater proportion of dry matter was distributed to the fruit, generating significantly higher yields. Therefore, the addition of urea to ammonium or nitrate promoted N metabolism and N utilization in cucumber plants, especially treatments with 50% $NO_3^-$-N + 50% $CO(NH_2)_2$, as the recommended nitrogen form in this study.

**Keywords:** nitrogen forms; cucumber; nitrogen metabolism; gene expression

## 1. Introduction

Cucumber (*Cucumis sativus* L.) is an important vegetable crop cultivated worldwide due to its economic and nutritional benefits. The growth and development of cucumber are associated with increased nitrogen dependence [1,2]. Nitrogen (N) is an important phytonutrient that is usually required in large amounts, and its deficiency generally limits plant growth and development [3,4]. Thus, N functions as a signaling element that sustains plant growth and development, particularly under stressful conditions [5,6]. Deficient or excessive amounts of N have a significant negative impact on vegetables [3,7], suggesting that changing the forms of nitrogen supply without changing the molar amount of nitrogen is a very important pathway that influences the growth of plants.

Plants have the ability to take up several chemical forms of nitrogen, and different nitrogen forms have significant effects on the growth of plants [8]. The most common forms of nitrogen are ammonium

($NH_4^+$), which has a positive charge; nitrate ($NO_3^-$), which has a negative charge; and urea ($CO(NH_2)_2$), which has no charge [9]. Due to its inexpensive price and high N content (46% of the mass), urea is the most common N fertilizer used in agriculture worldwide, accounting for approximately 50% of the total world N fertilizer consumption [10].

The three forms of nitrogen have a common metabolic pathway, the glutamate pathway, through which nitrogen is assimilated into amino acids, and glutamine synthase (GS) and glutamate synthesis (GOGAT) play roles; therefore, this pathway is also called the GS/GOGAT cycle [11]. $NH_3$ is converted into glutamine by GS, and glutamine is converted into glutamate by GOGAT; subsequently, these amino acids are assimilated into proteins, nucleic acids, etc. [12]. Glutamate dehydrogenase (GDH) plays an auxiliary role in the glutamate pathway by catalyzing the conversion of α-ketoglutaric acid and $NH_3$ into glutamate, and this enzyme can also catalyze the inverse reaction [13]. Therefore, GS, GOGAT, and GDH are regarded as the main enzymes of nitrogen metabolism in higher plants [14]. However, these three forms of nitrogen undergo different absorption and metabolic processes before they are assimilated into $NH_4^+$-N. Plants directly assimilate $NH_4^+$-N into glutamate and indirectly assimilate $NO_3^-$-N into glutamate by reducing nitrate to nitrite and then to $NH_4^+$-N and by converting urea to glutamate through urease (Ur), which catalyzes the hydrolysis of $CO(NH_2)_2$ to carbamate and $NH_3$ and then to $NH_4^+$-N [15,16]. Nitrate absorption is an active energy-requiring process involving an inverse electrochemical potential gradient [17]. There is an exclusive transport of nitrate nitrogen in the cytomembrane, in which $NO_3^-$ is transported into the cell by the proton driving force generated by plasma membrane hydrolysis [18], and $NH_4^+$-N uptake by plant roots involves the ammonium transporter family [19]. The $CO(NH_2)_2$ transport systems in plant root cells have been identified and are mediated by the urea transporter DUR3 and aquaporins [15].

In general, $NO_3^-$-N is considered the most effective nitrogen form for cucumber growth, and cucumber prefers $NO_3^-$-N to $NH_4^+$-N; thus, $KNO_3$ or $Ca(NO_3)_2$ is widely used in the production of cucumber [20–22]. However, scholars have also found that single nitrate could sometimes inhibit the growth of plants [23]. For good N nutrition, most plants prefer a mixture of $NO_3^-$ and $NH_4^+$ rather than $NH_4^+$ or $NO_3^-$ alone [15,24]. Single ammonium or nitrate nitrogen has adverse effects on the growth and development, yield, and quality of plants, while appropriate proportions of nitrate and ammonium nitrogen have favorable effects on plant growth, nitrogen absorption, and nitrogen metabolism processes, as well as on mineral element absorption [8,25]. Treatment with ammonium/nitrate at a ratio of 75/25 significantly increases the accumulation of N, P, and K in plants [26]. There are also reports on the effect of urea on plant growth. Combined applications of ammonium and nitrate were more effective than pure urea applications in perilla under salt stress [27]. On the other hand, a proper ratio of nitrate nitrogen and amide nitrogen can improve the nitrogen use efficiencies of pepper [28]. Additionally, urease activity was studied in soil [29] and potato [30]. Urea uptake was studied in maize [15] and oilseed [31]. However, few studies have reported on the metabolism and utilization of urea in crops, especially in cucumbers.

To date, there is limited information concerning the physiological and molecular responses of cucumber plants and the reciprocal influence of the different N forms. In this study, the responses to biochemical substances, N assimilation enzyme changes, nitrogen metabolism gene expression, and fruit yield of cucumber plants exposed to different N forms ($NH_4^+$-N, $NO_3^-$-N, $CO(NH_2)_2$, and their combinations) were examined. The purpose of these experiments is to determine whether urea and its combination are more favorable for cucumber growth and development and to explain its mechanism, further providing a theoretical basis for research on the nitrogen fertilizer supply for cucumber.

## 2. Materials and Methods

### 2.1. Plant Materials and Growth Conditions

The experiment was implemented in 2017 in a greenhouse at the Guizhou Academy of Agricultural Science (26°35′ N, 106°42′ E), China. The cucumber variety "Zhongnong 26" was acquired from Institute of Vegetables and Flowers, CAAS, P. R. China. The cucumber seeds were sterilized with 2.5% NaClO and germinated in gauze in the dark at 28 °C. After emergence, the seedlings were planted on plastic aperture disks (72 holes). Three weeks later, seedlings with two leaves were transplanted into plastic pots (diameter 25 cm and height 30 cm) containing 8 L of coir dust and grown in a greenhouse (28 ± 3 °C day/20 ± 2 °C night, relative humidity of 80%, 13 h light/11 h dark). There was one cucumber plant per pot, and their vines were hung when the cucumber plants had eight leaves. Cucumber plants were watered with nutrient solution at each watering to maintain the substrate water content at approximately 60% of its water-holding capacity. A tray was placed at the bottom of each pot to catch and return the leaked nutrient solution to the tray to fertilize the substrate in the pot.

### 2.2. Nitrogen Treatment

Each nitrogen treatment was characterized by the proportions of different nitrogen forms in the nutrient solution, and the nitrogen treatments were applied to each plant with the corresponding nutrient solution. Six treatments with the same total nitrogen contents were set with three single and three mixed forms of nitrogen (ammonium, nitrate, and amide) on the basis of Shanqi's formula [24], in which AN represents ammonium nitrogen, NN represents nitrate nitrogen, and UN represents amide (urea) nitrogen in the treatments (Table 1). These treatments were 14 mM N, which was $NH_4^+$ as $(NH_4)_2SO_4$ or $NH_4H_2PO_4$ and $NO_3^-$ as $Ca(NO_3)_2 \cdot 4H_2O$ or $KNO_3$. In all the N treatments, the other nutrient compositions were 3.5 Ca, 6.0 K, 2.0 Mg, and 1.0 P (mmol $L^{-1}$) and 110 Fe, 20.6 B, 0.16 Cu, 5.3 Mn, 0.49 Mo, and 0.34 Zn (μmol $L^{-1}$). For all N treatments, the solution pH was adjusted to 5.5 before addition to the substrate. Each experimental pot was fertilized with 1 L nutrient solution every five days until the harvest of the experimental plants; therefore, the applied nitrogen can be calculated from the nutrient solution input per cucumber plant throughout its growth period. In this experiment, there were two controls: one was 100% $NH_4^+$-N(AN) as a $NO_3^-$-N treatment with added $CO(NH_2)_2$, and the other was 100% $NO_3^-$-N(NN) as a $NO_3^-$-N treatment with added $CO(NH_2)_2$. Besides these, both 100% $CO(NH_2)_2$ and (50% $NH_4^+$ + 50% $NO_3^-$) were involved in the present study; the purpose of this was that the treatments could be compared with each other. Six treatments with 108 pots were employed in the experiment, allowing 18 pots for each treatment in three replicates. Six pots were randomly included in each replicate.

**Table 1.** The composition of nitrogen treatments in terms of different nitrogen forms.

| Treatments | Single Nitrogen Forms | Treatments | Mixed Nitrogen Forms |
|---|---|---|---|
| AN(Control 1) | 100% $NH_4^+$-N | AN-NN | 50% $NO_3^-$-N + 50% $NH_4^+$-N |
| NN(Control 2) | 100% $NO_3^-$-N | NN-UN | 50% $NO_3^-$-N + 50% $CO(NH_2)_2$ |
| UN | 100% $CO(NH_2)_2$ | AN-UN | 50% $NH_4^+$-N + 50% $CO(NH_2)_2$ |

AN: ammonium nitrogen; NN nitrate nitrogen; UN: urea nitrogen.

### 2.3. Determination of Plant Yield, Dry Weight, and Nitrogen Content

Measurement of yield: After the cucumber fruits reached harvest standard, which was usually 15 days after pollination, the fruits were harvested and weighed separately.

Determination of plant dry weight: The cucumber plants were harvested after undergoing N treatment for 60 days. At harvest, the plants were separated into leaf, stem, fruit, and root tissues before washing with distilled water. The fresh tissues were dried carefully at 105 °C for 20 min and then further dried at 80 °C for 48 h and subsequently weighed.

Nutrient element measurements: The dried samples were ground and passed through a 0.5 mm screen. Then, 0.5 g dry leaf, stem, fruit, and root tissue samples were soaked in 10 mL concentrated sulfuric acid ($H_2SO_4$) for 24 h, digested in 10 mL $H_2SO_4$ digestion buffer in a fume hood, and heated to 180 °C for 3 h, followed by the addition of 5 mL hydrogen peroxide ($H_2O_2$). The extract solution was transferred to 100 mL volumetric flasks and then diluted to 100 mL with deionized water for N assays. The N concentration was analyzed using an automatic Kjeldahl apparatus according to the manufacturer's instructions (K9840, Shanghai, China), and the nitrogen content was calculated [22].

### 2.4. Biochemical Analysis

The third top leaf at the 15-leaf stage was taken from cucumber plants, immediately frozen in liquid nitrogen, and stored at −80 °C for biochemical and expression analyses. The protein-dye binding method was used to determine the protein content and the chlorophyll relative quantity was measured using a SPAD-502 chlorophyll analyzer (Konica Minolta, Tokyo, Japan) [32]; nitrates and total free amino acids were measured by hydroxybenzoic acid colorimetric analysis and ninhydrin chromogenic analysis, respectively [33,34]; an ELISA method was used to determine glutamate; and the citrulline content was measured by an HPLC method as described by Witte [30,35].

The nitrate reductase activity and GS activity were determined by sulfanilamide colorimetry and γ-glutamyl hydroxamate analysis, respectively [33,36]; GOGAT activity was measured by HPLC method as described by Esposito [37]; Turano's method was used to determine GDH activity; and urease activity was measured by indophenol blue colorimetry [31,38].

### 2.5. Total RNA Isolation and cDNA Synthesis

The total RNA of the leaves from cucumber plants was extracted using a commercial RNA extraction kit (Tiangen, Beijing, China) according to the manufacturer's instructions. The RNA quality was then assessed using a One-Drop spectrophotometer (Nanodrop Technologies Inc., Montchanin, DE, USA). The total RNA was incubated at 42 °C for 2 min with gDNA Eraser (TaKaRa, Dalian, China) to remove genomic DNA contamination. cDNA was synthesized using the Prime Script RT Reagent Kit (TaKaRa, Dalian, China), according to the manufacturer's instructions. Finally, cDNA was diluted 15-fold with sterile $ddH_2O$, and 2 μL of this diluted cDNA was used for quantitative real-time PCR (qRT-PCR) analysis.

### 2.6. Gene Expression Analysis by qRT-PCR

To detect the expression level of N-metabolism-related genes in cucumber, a total of 18 genes involved in N metabolism were identified from the transcriptomic database (http://cucurbitgenomics. org/) of cucumber after alignment with the sequences in Arabidopsis (Table 2). Primers were designed using Primer Premier 5 software (Primer Co., Waterloo, ON, Canada) according to the acquired gene sequences. The primer sequences used for qRT-PCR are shown in Tables 2 and 3. qRT-PCR was carried out on the Bio-Rad CFX 96 Real-time System (Bio-Rad, San Diego, CA, USA) using TaKaRa SYBR Premix Ex Taq (TaKaRa, Dalian, China). Each PCR mixture comprised a total volume of 20 μL, containing 10 μL SYBR Premix Ex Taq, 7 μL sterile deionized water, 2 μL diluted cDNA template, 0.5 μL forward primer, and 0.5 μL reverse primer. qRT-PCR was conducted according to the manufacturer's instructions. The conditions were controlled as follows: 95 °C for 30 s, followed by 40 cycles at 95 °C for 10 s, 60 °C for 20 s, and 65 °C for 10 s. The internal parameter was Actin. The data were obtained by using Bubner's Ct ($2^{-\Delta\Delta Ct}$) method to calculate the relative gene expression [39].

**Table 2.** List of 18 genes related to nitrogen metabolism in cucumber.

| Gene Abbreviation | Accession No. | Biological Process and Molecular Function | Function |
|---|---|---|---|
| *NR-1* | XM004139616.2 | Cyclic pyranopterin monophosphate synthase accessory protein Mitochondrial transcript variant *X1*, | nitrate reduction |
| *NR-2* | NM001280767.1 | nitrate reductase [NADH]-like (*NR2*) | nitrate reduction |
| *NR-3* | XM004135853.2 | Cyclic pyranopterin monophosphate synthase, mitochondrial | nitrate reduction |
| *NR-4* | XM011661764.1 | Molybdopterin biosynthesis protein *CNX1* | nitrate reduction |
| *NiR* | XM004140647.2 | Ferredoxin-nitrite reductase, chloroplastic (*NIR*) | nitrite reduction |
| *GS-1* | NM001280715.1 | Glutamine synthetase cytosolic isozyme-like (*GS1*) | glutamine synthesis |
| *GS-2* | XM011661119.1 | Glutamine synthetase nodule isozyme-like | glutamine synthesis |
| *GS-3* | XM011656924.1 | Glutamine synthetase nodule isozyme | glutamine synthesis |
| *GS-4* | XM004134113.2 | Glutamine synthetase leaf isozyme, chloroplastic transcript variant *X1* | glutamine synthesis |
| *GOGAT-1-1* | XM004136730.2 | Ferredoxin-dependent glutamate synthase, chloroplastic, transcript variant *X1* | glutamate synthesis |
| *GOGAT-1-2* | XM011653889.1 | Ferredoxin-dependent glutamate synthase, chloroplastic, transcript variant *X2* | glutamate synthesis |
| *GOGAT-2-1* | XM011653296.1 | Glutamate synthase [NADH], amyloplastic, transcript variant *X1* | glutamate synthesis |
| *GOGAT-2-2* | XM011653298.1 | Glutamate synthase [NADH], amyloplastic, transcript variant *X3* | glutamate synthesis |
| *GDH-1* | XM004147487.2 | Glutamate dehydrogenase 1, transcript variant *X2* | glutamate dehydrogenation |
| *GDH-2* | XM004146845.2 | Glutamate dehydrogenase 2 | glutamate dehydrogenation |
| *GDH-3* | XM004143618.2 | Glutamate dehydrogenase 2-like | glutamate dehydrogenation |
| *Urease-1* | XM011657264.1 | Urease, transcript variant *X2* | urea decomposition |
| *Urease-2* | XM011659065.1 | Urease accessory protein *G* | urea decomposition |

**Table 3.** Primer sequences for qRT-PCRs.

| Gene | Forward Primer (5′) | Reverse Primer (3′) | Length |
|---|---|---|---|
| *NR-1* | TACTTCGGCTTTGACTCATGTTG | GTATGTTTTGCTCCGCTTATTCC | 207 bp |
| *NR-2* | TACTGGTGCTGGTGTTTCTGGTC | GATTTCTCCCTTGTGAGGTTTGC | 195 bp |
| *NR-3* | GAACTGTGTTATAATGCGTGGTT | CAAGTCGTGTAAGGTTTGTGAAG | 197 bp |
| *NR-4* | CAGAGAACACAGAAAAGAAGGAA | CACAATGAAACAGTGGTCGAATA | 207 bp |
| *NiR* | GTCCCTCTCTGTGGAGCCATCTT | CCCTTCTTTCCCATTGCTTATTT | 199 bp |
| *GS-1* | TTCTTTCTTTTGATCCAAAACCA | ATGTCGCCCTGTGAGACGACGCT | 197 bp |
| *GS-2* | CAAGTCGGTCCTACCGTTGGTATTG | TCGAAGTAGACCTGTAATTGGTG | 188 bp |
| *GS-3* | CTTTTGACCCCAAACCAATTCAG | GTGTCGACCAGTTAGACGACGCT | 191 bp |
| *GS-4* | GTGCCCATCCCTACAAACAAACG | ACACCACAGTAATAAGGCCCCTG | 185 bp |
| *GOGAT-1-1* | GAACGAGAACTTTACATTTGTAG | CTATATCTTCGATGATAAATAGC | 206 bp |
| *GOGAT-1-2* | GAAATTGATTGAAAGAGAAGCAA | CTATATCTTCGATGATAAATAGC | 183 bp |
| *GOGAT-2-1* | AGTTGGGATCGTGCTCAGCCT | CTAATTAAAAGCTCAAGAACACC | 216 bp |
| *GOGAT-2-2* | ATGCGTGTTTGGGCCACAATG | CTAATTAAAAGCTCAAGAACACC | 194 bp |
| *GDH-1* | GCAATCCTGGAGAATTAAGTATA | AGAGATCCACCTAGATCAATAGG | 217 bp |
| *GDH-2* | TAAAGAAAGTACTGGAAGCCTTG | TCATCTGCCTCTGGATCTGTGGG | 195 bp |
| *GDH-3* | ATTGATGTACCTGAGCTGATTATTCA | ATCTGCTTCTGGATCAGTAGGATG | 219 bp |
| *Urease-1* | GATGGCTTCATTATGACCCTTGG | TGTGATGCCGCTTGATATTGCTT | 195 bp |
| *Urease-2* | GAAAGGATTAGAGCAGTGGAAAC | AGACACATCGATGATATAGATGA | 201 bp |

## *2.7. Statistical Analysis*

The analysis of the variance was conducted by way of ANOVA with Tukey's multiple range test at $p < 0.05$ in different N forms. Analyses were conducted on three replicates per treatment. Pearson's correlation was carried out using SPSS 22.0 software (version 22, International Business Machines Corporation, Armonk, NY, USA).

## 3. Results

### *3.1. Effect of Different Nitrogen Forms on Cucumber Growth*

The AN treatment showed the minimum dry matter (Figure 1A,C), nitrogen content (Figure 1D), and yield (Figure 1E), and this treatment was significantly lower than the other nitrogen treatments. The NN-UN treatment exhibited the maximum dry matter weights in plants, leaves, and fruits among

the six treatments (Figure 1A,C). Furthermore, analysis of the dry matter proportion of the fruit weight to the total weight in the six treatments revealed a greater proportion of dry matter distributed to the fruit in the NN-UN treatment than in other treatments and a greater proportion of dry matter in the fruit after AN-UN treatment than after AN treatments (Figure 1A,F). The nitrogen content of the plants was highest after mixed or single urea treatments (UN, NN-UN, AN-UN), followed by the NN and AN-NN treatments (Figure 1D). These results showed that the nitrogen content in treatments containing urea was higher than that in treatments without urea (Figure 1B). The yield was highest in treatment with NN-UN, followed by AN-UN, UN, NN, and AN-NN treatments (Figure 1E), which indicated that treatment with $NO_3^--N$ and the addition of equal $CO(NH_2)_2$ could improve the yield of cucumber. Additionally, $NH_4^+-N$ treatment with added equal $CO(NH_2)_2$ also improved the dry matter weight, nitrogen content, and yield of cucumber.

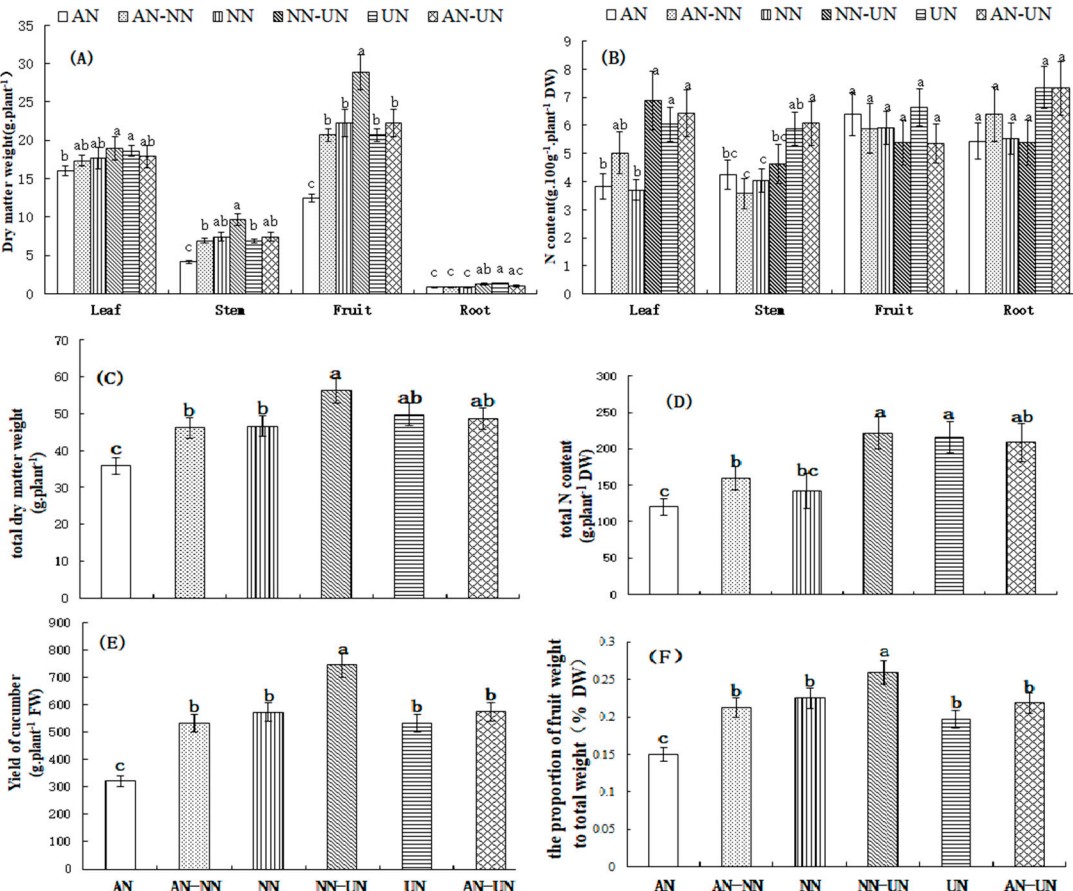

**Figure 1.** Effect of different nitrogen forms on dry matter weight, nitrogen content, and yield in cucumber. (**A**) Dry matter weight of each individual organ; (**B**) Nitrogen content of each individual organ; (**C**) Dry matter weight of each individual plant; (**D**) Nitrogen content of each individual plant; (**E**) Yield of each individual plant; (**F**) Effect on root–shoot ratio of the nitrogen form. Bars represent the standard error. Different letters above the bars indicate significant differences at $p < 0.05$ according to Tukey's multiple comparison test.

*3.2. Effect of Different Nitrogen Forms on the Biochemical Metabolites in Cucumber*

The highest content of proteins, free amino acids, and glutamate was observed in the NN-UN treatment and the lowest content was observed in the AN treatment, while a moderate content was detected in the other treatments (Figure 2A–C). The results demonstrated different protein, amino acid, and glutamate contents in cucumber after treatment with different nitrogen forms, and the trend was similar to that of the dry matter and yield.

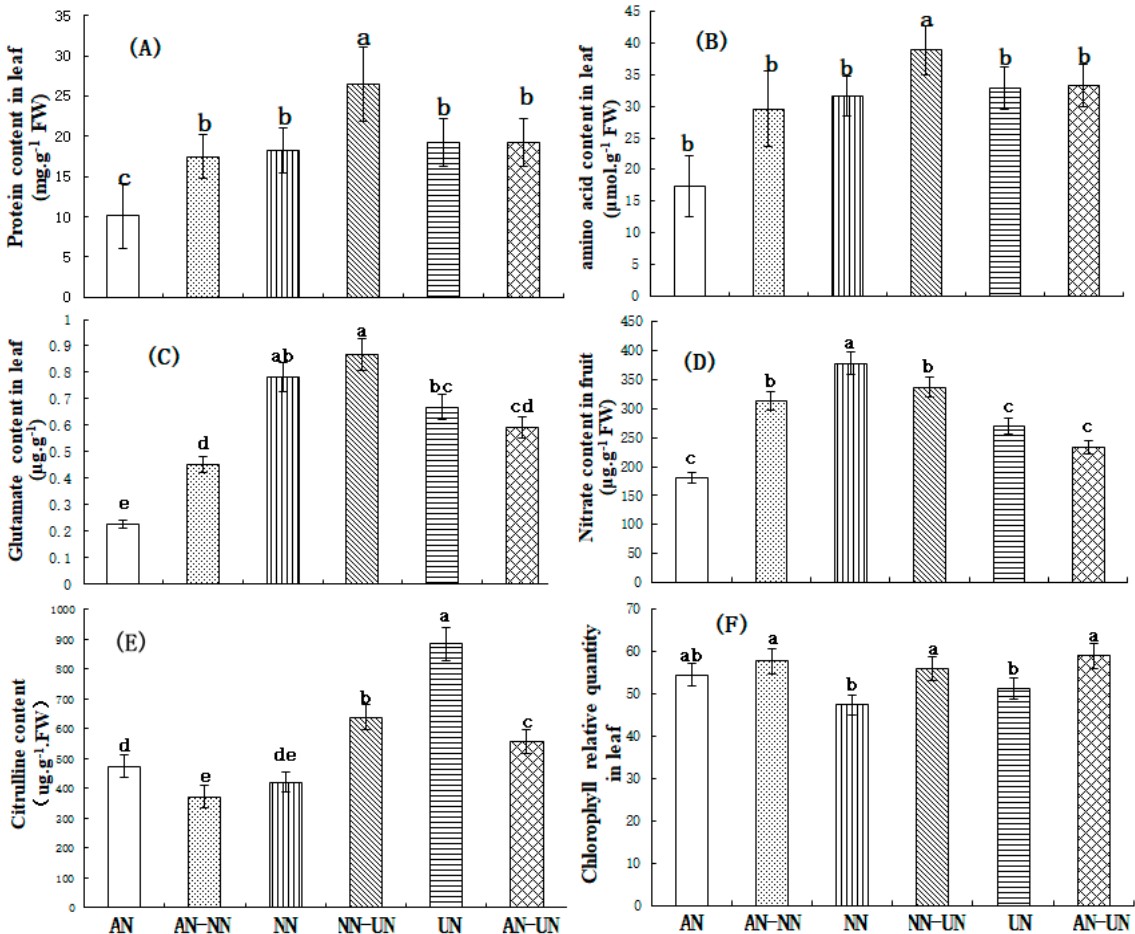

**Figure 2.** Effect of different nitrogen forms on the biochemical metabolites in cucumber. (**A**) protein content; (**B**) Amino acid content; (**C**) Glutamate content; (**D**) Nitrate content; (**E**) Citrulline content; (**F**) chlorophyll relative content. Bars represent the standard error. Different letters above the bars indicate significant differences at $p < 0.05$ according to Tukey's multiple comparison test.

The AN treatment showed the lowest nitrate content and glutamate content (Figure 2C,D). The glutamate content was highest after the NN-UN treatment, followed by the NN and UN treatments (Figure 2C). This trend was similar to that of the dry matter and yield of cucumber. Also, the NN treatment exhibited the maximum nitrate content, and the mixed $NO_3^-$ treatments (AN-NN, NN-UN) manifested the second highest nitrate content (Figure 2D). The results showed that the application of $NO_3^-$-N increased the content of nitrate in cucumber. The nitrate content in cucumber can be reduced by changing the form of nitrogen under application of the same amount of nitrogen.

The UN treatment manifested the maximum citrulline content and the NN-UN treatment manifested the second highest citrulline content, while the AN-NN treatment showed minimum citrulline content and the AN treatment showed the second lowest citrulline content (Figure 2E). These results showed that the citrulline content in the treatment with added $CO(NH_2)_2$ was higher than that in the treatment with added $NH_4^+$-N with equal proportions of $NO_3^-$-N, in which the content of the former was twice as much as that of the latter. In addition, the citrulline content in treatments containing $CO(NH_2)_2$ was higher than that in treatments without $CO(NH_2)_2$, while that in the AN-UN treatment was higher than that in the AN treatment (Figure 2E).

The chlorophyll relative content of the two mixed nitrogen treatments (AN-NN, NN-UN) was higher than that of the single nitrogen application (AN, NN, UN). The mixed $NH_4^+$ treatment (AN-UN, AN-NN) showed the highest chlorophyll relative content, while the NN treatment manifested the lowest chlorophyll relative content (Figure 2F).

### 3.3. The Correlation between Biochemical Substance and the Growth Index of Cucumber

The leaf protein content and amino acid content were significantly correlated with leaf dry matter and total dry matter; their correlation coefficients were 0.930, 0.954, 0.968, and 0.985, respectively. Moreover, there was a significant positive correlation between leaf protein content and yield ($r = 0.986$), revealing that different nitrogen forms affected the protein content of cucumber and that cucumber yield is inseparable from its leaf protein content (Table 4).

**Table 4.** Correlations between different parameters.

| Correlation Coefficient (r) | Leaf Dry Matter (g·plant$^{-1}$) | Total Dry Matte (g·plant$^{-1}$) | Leaf N Content (g·plant$^{-1}$) | Leaf Protein Content (g·plant$^{-1}$) | Total N Content (g·plant$^{-1}$) | Nitrate (µg·g$^{-1}$) |
|---|---|---|---|---|---|---|
| Leaf dry matter (g·plant$^{-1}$) | – | 0.987 ** | – | – | – | – |
| Leaf N content (g·plant$^{-1}$) | 0.895 * | 0.861 * | – | – | – | – |
| Leaf protein content (mg·g$^{-1}$) | 0.930 ** | 0.968 ** | 0.768 | – | – | – |
| Leaf amino acid content (µmol·g$^{-1}$) | 0.954 ** | 0.985 ** | 0.741 | 0.967 ** | 0.827 * | – |
| Total N content (g·plant$^{-1}$) | 0.895 * | 0.888 * | 0.962 ** | 0.813 * | – | – |
| Yield (g·plant$^{-1}$) | 0.885 * | 0.939 ** | 0.684 | 0.986 ** | 0.731 | – |
| NR (µmol·h$^{-1}$·g$^{-1}$) | – | – | – | – | – | 0.871 * |

* Indicates significance at the 0.05 probability levels. ** Indicates significance at the 0.01 probability levels.

There was also a significant positive correlation between the N content and the dry matter of leaves, as well as between the total N content and total dry matter ($r = 0.930, 0.888$), indicating that the N content had a close relationship with dry matter under the same application amount but with different forms of nitrogen supply (Table 4); thus, the yield and dry matter weight of cucumber can be increased by changing the form of nitrogen to improve the nitrogen content under the same amount of nitrogen supply.

Additionally, correlation analysis showed that there are significant positive correlations between NR activity and nitrate in cucumber ($r = 0.871$) (Table 4). This result indicates that the NR activity may be enhanced when the proportion of $NO_3^-$-N in treatments is increased.

### 3.4. Effect of Different Nitrogen Forms on Nitrogen-Metabolizing Enzymes in Cucumber

The NN-UN treatment exhibited maximum GOGAT activity and GS activity in the leaves among the six treatments with equal nitrogen (Figure 3A,B). Whether $NH_4^+$-N, $NO_3^-$-N, or $CO(NH_2)_2$, mixed nitrogen (AN-NN, NN-UN, AN-UN) showed higher GOGAT and GS activity than single nitrogen (AN, NN, UN). Conversely, single nitrogen showed higher GDH activity than mixed nitrogen, especially the AN treatment, which exhibited the highest activity (Figure 3C).

The NN-UN treatment exhibited the highest NR activity, and the AN-NN and NN treatment containing $NO_3^-$ manifested the second highest NR activity, while other treatments without $NO_3^-$-N showed minimum NR activity (Figure 3D). These results showed that NR activity was improved in the $NO_3^-$-N treatment with added $CO(NH_2)_2$. Similarly, the urease activity was highest in the single urea treatment (UN), followed by the mixed urea treatments (NN-UN, AN-UN), and the other treatments without urea showed the lowest activity (Figure 3E). This result indicated that urease activity may be induced by the application of $CO(NH_2)_2$.

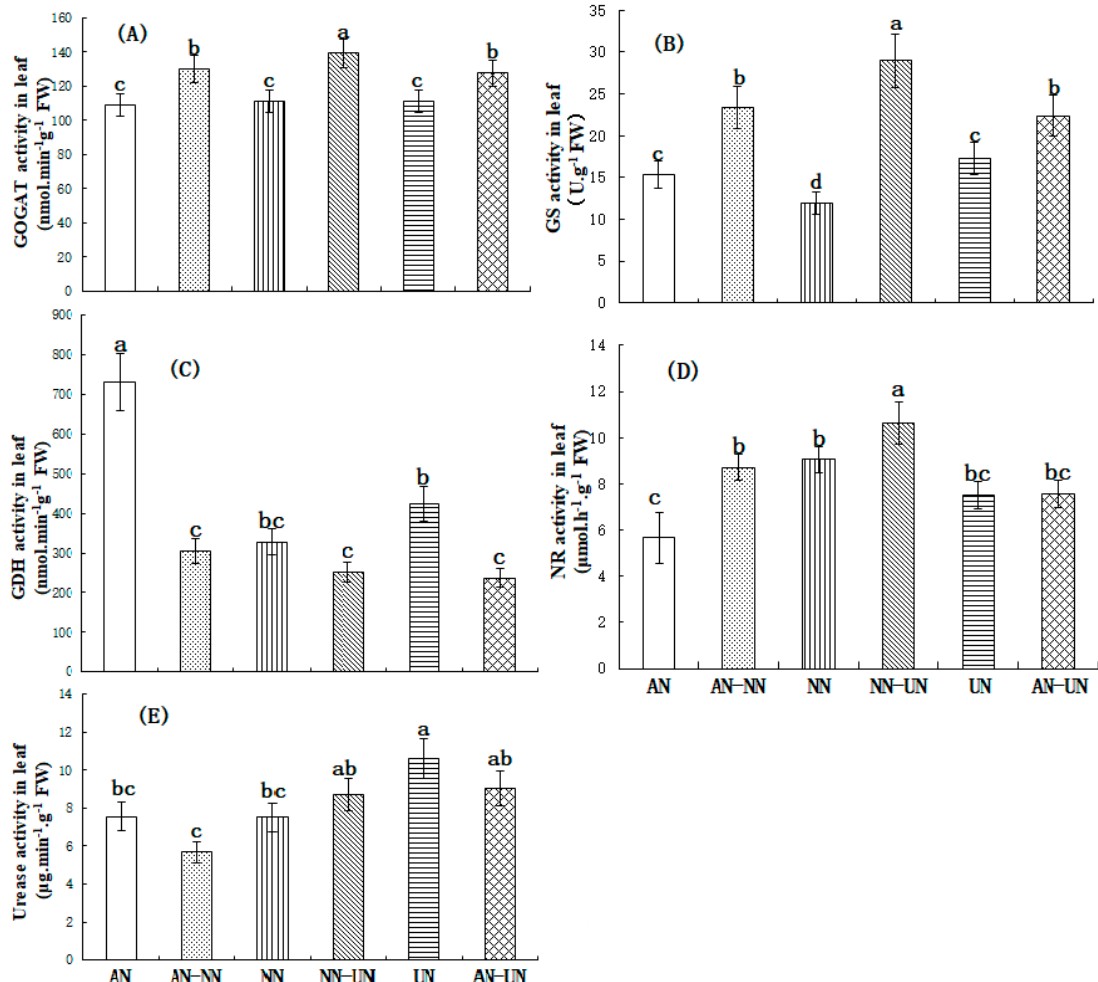

**Figure 3.** Effect of different nitrogen forms on nitrogen-metabolizing enzymes in cucumber. (**A**) GOGAT activity; (**B**) GS activity; (**C**) GDH activity; (**D**) NR activity; (**E**) Urease activity. Bars represent the standard error. Different letters above the bars indicate significant differences at $p < 0.05$ according to Tukey's multiple comparison test.

*3.5. Expression Profiles of the N-Metabolism-Related Genes of Cucumber Treated with Different Nitrogen Forms*

qRT-PCR was used to measure the relative expression levels of 18 genes involved in the nitrogen metabolism pathway in cucumber leaves under different nitrogen treatments (Table 2). In this experiment, the gene expression level was obtained for each gene on the basis of delta Ct at each treatment. Figure 4B,C show that the expression levels of the two genes involved in nitrate reduction, *NR-2* and *NR-3*, were increased after $NO_3^-$-N treatment with added $NH_4^+$-N or $CO(NH_2)_2$ (NN-UN, AN-NN). The expression levels of *NiR*, the gene involved in nitrite reduction, increased after $NO_3^-$-N treatment with added $NH_4^+$-N or $CO(NH_2)_2$ (NN-UN, AN-NN), showing the same tendency as *NR-2* and *NR-3* (Figure 4E). The expression levels of the *UR-2* gene involved in the urea cycle were increased significantly after $NO_3^-$-N treatment with added $CO(NH_2)_2$ (NN-UN), and its expression levels were also increased after $NH_4^+$-N treatment with added $CO(NH_2)_2$ (AN-UN). However, the expression levels of the *UR-1* gene could not be detected in the experiment (Figure 4F).

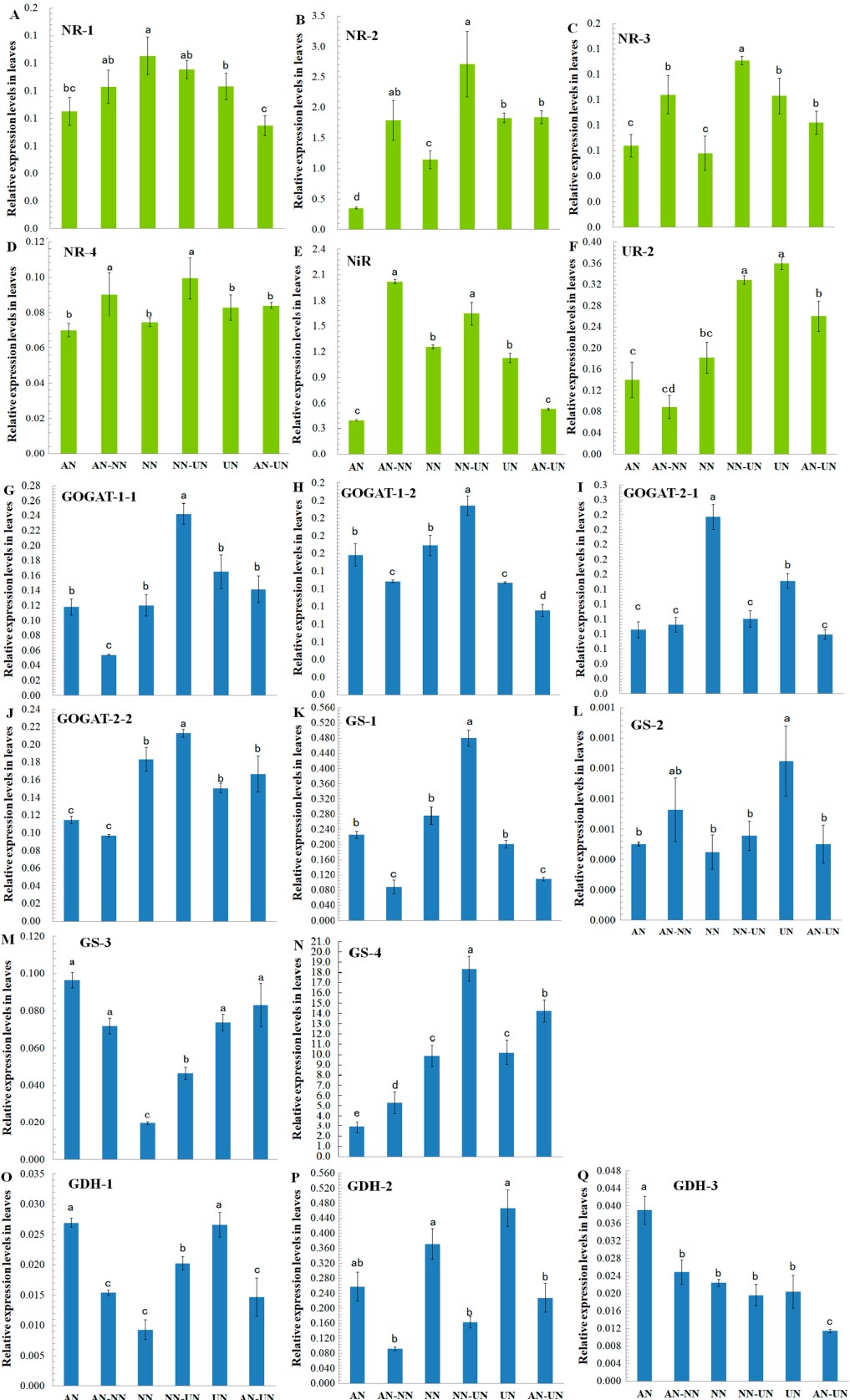

**Figure 4.** The expression profiles of genes involved in metabolizing nitrogen in cucumber. (**A**) NR-1; (**B**) NR-2; (**C**) NR-3; (**D**) NR-4; (**E**) NiR; (**F**) UR-2; (**G**) GOGAT-1-1; (**H**) GOGAT-1-2; (**I**) GOGAT-2-1; (**J**) GOGAT-2-2; (**K**) GS-1; (**L**) GS-2; (**M**) GS-3; (**N**) GS-4; (**O**) GDH-1; (**P**) GDH-2; (**Q**) GDH-3. Bars represent the standard error. Different letters above the bars indicate significant differences at *p* < 0.05 according to Tukey's multiple comparison test.

There are 11 genes involved in the glutamate cycle of nitrogen metabolism in cucumber, including the glutamate synthesis genes *GOGAT-1-1*, *GOGAT-1-2*, *GOGAT-2-1*, and *GOGAT-2-2*; the glutamine synthesis genes *GS-1 GS-2*, *GS-3*, and *GS-4*; and the glutamate dehydrogenation genes *GDH-1*, *GDH-2*, and *GDH-3* (Figure 4G–Q). Figure 4G shows that the expression level of the glutamate synthesis gene *GOGAT-1-1* was increased after $NO_3^-$-N treatment with added $CO(NH_2)_2$ (NN-UN), while its expression levels decreased after $NO_3^-$-N treatment with added $NH_4^+$-N (AN-NN). In addition, the expression levels of the *GS-1* and *GS-4* genes involved in glutamine synthesis were increased after $NO_3^-$-N treatment with added $CO(NH_2)_2$, while their expression levels decreased after $NO_3^-$-N treatment with added $NH_4^+$-N. The expression levels of the two genes *GS-4* and *GOGAT-2-2* increased after $NH_4^+$-N treatment with added $CO(NH_2)_2$ (Figure 4K,L). The glutamate dehydrogenase gene *GDH-3* had higher expression levels after treatment with single $NH_4^+$-N (AN) than after other treatments. Additionally, the *GDH-2* gene had a higher expression level with a single nitrogen form than with mixed nitrogen forms (Figure 4P).

## 4. Discussion

### 4.1. Nitrogen Assimilation Response to Nitrogen Forms Prior to the Glutamate Cycle

Nitrogen assimilation mainly includes the reduction of $NO_3^-$-N and hydrolysis of $CO(NH_2)_2$ before nitrogen enters the glutamate cycle. In general, $NO_3^-$-N is thought to be the most effective nitrogen form for cucumber [21,22], but the application of single nitrate easily leads to excessive nitrate content in spinach, and the yield is sometimes not ideal [40]. A study has shown that the nitrate content in Chinese cabbage is too high, and the quality of vegetable is affected when large amounts of nitrate nitrogen are absorbed [41]. Therefore, determining how to promote nitrate reduction and reduce nitrate content is important. The NR activity of the single $NO_3^-$-N treatment was lower than that of the $NO_3^-$-N treatment with added $CO(NH_2)_2$ in our study, which showed that the $NO_3^-$-N treatment with added equal $CO(NH_2)_2$ promoted nitrate reduction in cucumber. A similar result showed that the content of nitrate in Chinese cabbage was reduced when $NO_3^-$-N was partially replaced with $CO(NH_2)_2$ [42]. However, the molecular responses of $CO(NH_2)_2$ were not investigated in their study. Our study found that the gene expression levels of the *NR-2*, *NR-3*, and *NiR* genes involved in nitrate reduction were increased after $NO_3^-$-N treatment with added $NH_4^+$-N or $CO(NH_2)_2$, which could promote nitrate reduction and reduce the nitrate content of cucumber.

At present, few studies examine urease activity, *UR* gene expression, and $CO(NH_2)_2$ metabolism. In the present study, we found that the urease activity was highest in single urea treatments, which indicated that this activity may be induced by the application of $CO(NH_2)_2$. The expression levels of the *UR-2* gene involved in urea degradation were increased significantly after $NO_3^-$-N treatment with added $CO(NH_2)_2$. Additionally, the expression levels of the *UR-2* gene were increased after $NH_4^+$-N treatment with added $CO(NH_2)_2$. Moreover, dry matter and yield in the $NO_3^-$-N and $CO(NH_2)_2$ treatments were higher than those in the $NO_3^-$-N and $NH_4^+$-N treatments with equal nitrogen in the two mixed $NO_3^-$-N treatments, which may be due to the multi-pathway metabolism of $CO(NH_2)_2$. Urea metabolism mainly occurs through two pathways: the urease pathway and the amino acid pathway. In the urease pathway, urea is hydrolyzed to $NH_3$ and then enters into the GS/GOGAT pathway, which is the same pathway as that for ammonium assimilation [43,44]. The amino acid metabolic pathway is completely different from the ammonium and nitrate metabolism pathway in that urea is not decomposed to $NH_3$ but is assimilated directly into amino acids. First, $CO(NH_2)_2$ is transferred into the cell by transport molecules and combines with phosphate groups; then, this molecule is synthesized into citrulline. Subsequently, citrulline enters the urea cycle and synthesizes the amino acids that plants need. Thus, citrulline is the most important intermediate in $CO(NH_2)_2$ metabolism, and its content reflects the activity of amide nitrogen metabolism in the amino acid pathway [31,44]. In the present study, the citrulline content after treatment with added $CO(NH_2)_2$ was higher than that after treatment with added $NH_4^+$-N with equal proportions of $NO_3^-$-N, in which the content of the

former is twice as much as that of the latter. Therefore, we deduced that the amino acid pathway of urea metabolism promoted N metabolism in treatments containing $CO(NH_2)_2$ and enhanced the protein content and N content in the leaves, which further increased the dry matter weight of plants. Similarly, the amino acid pathway metabolism of $CO(NH_2)_2$ may cause higher dry matter weight and yield after $NH_4^+$-N and $CO(NH_2)_2$ treatments than after the pure $NH_4^+$-N treatment.

After $NO_3^-$-N treatment with added equal $CO(NH_2)_2$, the expression levels of the *NR-2*, *NR-3*, and *NiR* genes involved in nitrate reduction were increased, and the NR activity increased, promoting nitrate assimilation. Moreover, the mixture of $NO_3^-$-N and $CO(NH_2)_2$ exhibited higher urease activity and increased the expression levels of the *UR-2* gene involved in urea metabolism, resulting in the promotion of urea metabolism. In addition, urea metabolism lies in the amino acid metabolic pathway to enhance metabolic channels; urea is not decomposed to $NH_3$ but is assimilated directly into amino acids. Consequently, the $NO_3^-$-N treatment with added $CO(NH_2)_2$ promoted nitrogen metabolism to improve biomass but reduced the content of nitrate. After $NH_4^+$-N treatment with added equal $CO(NH_2)_2$, the change in nitrogen form promoted both the urease pathway and the amino acid metabolic pathway of urea metabolism, resulting in the partial substitution of $CO(NH_2)_2$ with $NH_4^+$-N, which also promoted nitrogen metabolism to improve biomass.

### 4.2. Glutamate Cycle Response to Nitrogen Forms

The three forms of nitrogen have a common metabolic pathway, the glutamate pathway, in which $NH_4^+$-N is assimilated into amino acids, and three important enzymes—GS, GOGAT, and GDH—play a role (Figure 1). The present study showed that the mixed form of nitrogen had higher GOGAT activity and GS activity than the single form of nitrogen, and the mixed $NO_3^-$-N and $CO(NH_2)_2$ supply exhibited the maximum GOGAT activity and GS activity in plant leaves among the six treatments with equal nitrogen. The results suggest that mixed nitrogen may promote nitrogen metabolism. Similar conclusions have been drawn from studies on the effects of N forms on GS/GOGAT after $NO_3^-$-N treatment with added $NH_4^+$-N in cucumber [45] and cotton seedlings [46], but $CO(NH_2)_2$ was not included in their study. Furthermore, we found that the gene expression levels were slightly different from enzyme activity but consistent with dry matter accumulation. The expression levels of the genes *GOGAT-1-1*, *GS-1*, and *GS-4* involved in the glutamate cycle were increased after $NO_3^-$-N treatment with added $CO(NH_2)_2$, while the expression levels of the former two genes decreased after $NO_3^-$-N treatment with added $NH_4^+$-N. This finding suggested that $NO_3^-$-N treatment with added equal $CO(NH_2)_2$ could promote nitrogen metabolism, but $NO_3^-$-N treatment with added equal $NH_4^+$-N could restrain nitrogen metabolism. Moreover, the $NH_4^+$-N treatment with added equal $CO(NH_2)_2$ also promoted nitrogen metabolism, and the expression levels of the two genes related to nitrogen metabolism (*GS-4*, *GOGAT-2-2*) were increased. In addition, the change in glutamate content was similar to that of dry matter and cucumber yield in the present study because glutamate was the most important intermediate in the nitrogen metabolism pathway, and its content reflects the vitality of nitrogen metabolism [35]. After $NH_4^+$-N treatment with added equal $CO(NH_2)_2$, the glutamate content in cucumber was enhanced.

Contrary to GS/GOGAT, a single form of nitrogen manifested higher GDH activity than mixed nitrogen, and the single $NH_4^+$-N treatment manifested the highest activity, which indicated that GDH played a critical role as the GS-GOGAT cycle was restrained. Many studies reported that the role of GDH in the nitrogen metabolism pathway was greatly increased during plant growth stress, especially ammonia toxicity, and suggested that GDH may play an important role in removing excessive ammonia and removing ammonia toxicity in plants [47,48]. Moreover, gene expression analyses in the present study also indicated GDH activity. *GDH-3* had higher expression levels in the treatment with single $NH_4^+$-N than in other treatments. Additionally, the *GDH-2* gene with a single form of nitrogen showed higher expression levels than with the mixed form of nitrogen, which was consistent with GDH activity. Additionally, the deamination catalyzed by GDH can provide a sufficient carbon skeleton for the tricarboxylic cycle in plants under stress [13]. In the present study, there was higher nitrogen content,

dry matter weight, and yield in the treatment with 50% $NH_4^+$-N + 50% $CO(NH_2)_2$ than in the 100% $NH_4^+$-N treatment; this may be because the substitute of $CO(NH_2)_2$ for $NH_4^+$-N relieves the ammonia toxicity of the pure $NH_4^+$-N treatment, which downregulates the *GDH-3* gene expression response to the changes after $NH_4^+$-N treatment with added $CO(NH_2)_2$.

As a result, the $NO_3^-$-N treatment with added equal $CO(NH_2)_2$ increased *GOGAT-1-1* and *GS-4* gene expression, improved GOGAT and GS activities, promoted the GS-GOGAT cycle of nitrogen metabolism, and increased the protein and nitrogen contents in cucumber leaves. Additionally, the treatment of $NH_4^+$-N with added equal $CO(NH_2)_2$ increased *GOGAT-2-2* and *GS-4* gene expression, improved GOGAT and GS activities, promoted the GS-GOGAT cycle, increased the protein and nitrogen contents in cucumber leaves, and relieved the ammonia toxicity of pure $NH_4^+$-N treatment.

*4.3. Utilization of the Nitrogen Response to Nitrogen Forms*

From the experimental findings about nitrate reduction, urea hydrolysis, and $NH_4^+$-N assimilation of cucumber, we can observe that the majority of mixed nitrogen could promote nitrogen metabolism, including $NO_3^-$-N with added equal $CO(NH_2)_2$, $NH_4^+$-N with added equal $CO(NH_2)_2$, and $NH_4^+$-N with added equal $NO_3^-$-N. The mixed forms of nitrogen had higher GOGAT activity and GS activity than the single forms of nitrogen, which suggested that mixed nitrogen may promote the GS-GOGAT cycle of nitrogen metabolism. Additionally, the chlorophyll relative content in plants after the three mixed nitrogen treatments was higher than that after the single nitrogen application, which indicated that mixed nitrogen treatments with high chlorophyll relative content could promote photosynthesis in cucumber.

In experiments on cucumber growth, both $NO_3^-$-N treatment with added equal $CO(NH_2)_2$ and $NH_4^+$-N treatment with added $CO(NH_2)_2$ could improve dry matter weight and total nitrogen content, and more dry matter was distributed to the fruit, resulting in a significantly higher yield of cucumber than after treatment with pure $NO_3^-$-N or pure $NH_4^+$-N. However, not all mixed nitrogen forms of equal ratio could enhance dry matter weight and yield. The $NO_3^-$-N treatment with added equal $NH_4^+$-N did not improve the total dry matter weight or yield in the experiment. What accounts for this finding? In addition to being affected by the expression of nitrogen metabolism genes involved in nitrate reduction, urea utilization, and the glutamate cycle, $NO_3^-$-N treatment with added equal $NH_4^+$-N may lead to ammonia toxicity. Scholars believe that high ammonium concentrations can inhibit the absorption of $K^+$ and $Ca^{2+}$ through ammonia toxicity [43,49]. The $NH_4^+$-N + $NO_3^-$-N treatment has been reported to affect the absorption and assimilation of nitrogen because excessive ammonium in the nutrient solution led to toxic effects on muskmelon [50]. The growth of lettuce, spinach, cabbage, and garden sass were good with a single $NO_3^-$-N source, while the growth of these four vegetables was restrained with a $NO_3^-$-N + $NH_4^+$-N (1:1) N source and decreased greatly with single $NH_4^+$-N [51]. Similar conclusions about ammonia toxicity have also been shown by other scholars in studies on nitrogen utilization in cucumbers [52,53]. The present study found that pure $NH_4^+$-N treatment had a minimum effect on the content of glutamate, protein, and nitrogen; dry matter; and yield compared with the effects of the other treatments, which may be due to the ammonia toxicity. $NH_4^+$-N treatment with added equal $CO(NH_2)_2$ is likely to significantly increase the total dry matter weight and may relieve the ammonia toxicity of pure $NH_4^+$-N treatment.

To reveal the effects of biochemical metabolism on cucumber growth, the important nitrogen metabolites were studied herein. The results showed that leaf protein content was significantly correlated with dry matter and yield. In line with these relationships, we deduced that leaf protein content is inseparable from the yield of cucumber under treatment with different nitrogen forms but with the same amount of nitrogen. Other studies have also reported that the biochemical processes of plants are also different with different nitrogen forms [54]. Furthermore, there was a significant positive correlation between leaf nitrogen content and leaf dry matter and between total nitrogen content and total dry matter in the present study. Therefore, it seems feasible that the dry matter

weight of cucumber could be increased by changing the form of nitrogen to improve the nitrogen content under conditions with the same amount of nitrogen.

## 5. Conclusions

In the present study, the expression levels of six genes, including *US-2*, *NR-2*, *NR-3*, *NiR*, *GOGAT-1-1*, and *GS-4*, were upregulated after treatment with $NO_3^-$-N with added equal $CO(NH_2)_2$, demonstrating that urea addition could increase the expression of the genes involved in nitrogen metabolism. The gene expression levels of *US-2*, *GOGAT-2-2*, and *GS-4* were upregulated after treatment with $NH_4^+$-N with added equal $CO(NH_2)_2$, but GDH-3 expression was downregulated, demonstrating that urea could increase the expression of nitrogen metabolism genes and relieve the ammonia toxicity of pure $NH_4^+$-N treatment. Consequently, treatment with both $NO_3^-$-N with added equal $CO(NH_2)_2$ and $NH_4^+$-N with added $CO(NH_2)_2$ enhanced GOGAT, GS, and UR activities; increased the amino acid pathway of N metabolism; improved the content of glutamate, protein, chlorophyll, and nitrogen; and improved the dry matter weight. Furthermore, a greater proportion of dry matter was distributed to the fruit to improve yield. Additionally, the mixed $NO_3^-$-N and $CO(NH_2)_2$ supply promoted nitrate reduction, urea utilization, and the GS-GOGAT cycle of nitrogen metabolism in cucumber plants but reduced the nitrate content in cucumber. Therefore, treatment with 50% $NO_3^-$-N + 50% $CO(NH_2)_2$ was the most appropriate combination of nitrogen forms used in the study. Also, it must be pointed out that the result cannot be used in soil culture by farmers because this experiment was carried out in coir dust rather than in soil, but it could provide a scientific reference from a fundamental and plant physiology point of view in soilless culture. Additionally, a significant positive correlation was found between nitrogen content and dry matter as well as between leaf protein content and yield in different treatment forms with equal nitrogen, indicating that different nitrogen forms affected the protein content in the leaves, which was correlated with the yield and dry matter of cucumber.

**Author Contributions:** W.J. conceived and designed the experiment; C.M., T.B. and X.L. performed the experiments; H.Y. and Q.L. analyzed the data; C.M. wrote the paper; W.J., C.M. and J.X. revised the paper. All authors have read and approved the final manuscript.

**Funding:** This study was supported by the National Natural Science Foundation of China [31760596] and the Youth Fund of Guizhou Academy of Agricultural Science [2018(84)].

**Acknowledgments:** We thank Heng Wang for gene searching. We thank Tiantian Ban and Xiaohui Li for their assistance. Finally, we acknowledge for their helpful discussions all the members of our research team. In addition, we thank American Journal Experts (www.aje.com) for its linguistic assistance during the preparation of this manuscript.

**Conflicts of Interest:** The authors declare that they have no conflict of interest.

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
