# Peer review of "Urea Addition Promotes the Metabolism and Utilization of Nitrogen in Cucumber"

_agronomy, doi:10.3390/agronomy9050262_

Reviewer 1 Report

The study investigated nitrogen metabolism in cucumber and found that urea addition could improve N metabolism and utilization. It must be considered that this experiment was carried out in coir dust rather than soil, therefore, the result can not be used by farmers as mentioned in the Conclusion section. Soil microbial community including urease producing microbes or nitrifiers, will change nitrogen forms in soil and results will differ in different soils depends upon soil microbes. But the research is very good from fundamental and plant physiology point of view.

English needs to be improved, particularly in Materials and Methods section. Some methods need to be changed to be concise and clear. For example, Glutamate was measured by... Nitrate was measured.......Same sentences for each metabolite.

This kind of writing is long and not appropriate, you can combine them, Different methods were used to measure the biochemical metabolites of our interest; glutamate by... nitrate by....

You can also improve the Result section, I see that the most of results obey from a similar pattern, for example; in the most plant growth factors, treatment AN is minimum, you do not need to mention it separately for each factor: yield in cucumber for AN was minimum, total N concentration for AN was minimum...You can combine most of the results to make the text better.

There are some grammatical problems, please fix them. "content" has been written two times in Fig. 2D.

Not sure about the number of pots, if one pot is considered as one replicate, you will have three pots for each treatment while you have mentioned 18, it is confusing. Did you have pseudoreplicate?

In Methods section there is no specific explanation regarding Citrulline measurement. You need to mention which amino acids you measured.

Author Response

Dear reviewer

  Firstly, I would like to express our sincere thanks to you for the constructive and positive comments.

Point 1: The study investigated nitrogen metabolism in cucumber and found that urea addition could improve N metabolism and utilization. It must be considered that this experiment was carried out in coir dust rather than soil, therefore, the result can not be used by farmers as mentioned in the Conclusion section. Soil microbial community including urease producing microbes or nitrifiers, will change nitrogen forms in soil and results will differ in different soils depends upon soil microbes. But the research is very good from fundamental and plant physiology point of view.

Response 1: Thank you for your good suggestion. Indeed, our study was based on soilless cultivation rather than soil, and we have revised the conclusion section in revised manuscript (The revised details can be found in Line 450-453, page 15). “Also, it must be point out that the result can not be used in soil culture by farmers because this experiment was carried out in coir dust rather than soil, but it could provide a scientific reference for fundamental and plant physiology point of view in soilless culture.”

Point 2: English needs to be improved, particularly in Materials and Methods section. Some methods need to be changed to be concise and clear. For example, Glutamate was measured by... Nitrate was measured.......Same sentences for each metabolite. This kind of writing is long and not appropriate, you can combine them, Different methods were used to measure the biochemical metabolites of our interest; glutamate by... nitrate by....

Response 3: It is true that partial statement was not concise in the manuscript, the statement of materials and methods section have been changed in revised manuscript (The revised details can be found in Line 140-149, page 4). Besides, some sentences of other sections in the manuscript have been modified and marked with yellow background as well.

Point 3: You can also improve the Result section, I see that the most of results obey from a similar pattern, for example; in the most plant growth factors, treatment AN is minimum, you do not need to mention it separately for each factor: yield in cucumber for AN was minimum, total N concentration for AN was minimum...You can combine most of the results to make the text better.

Response 3: Your advice is very important for us, we have found the shortcomings in current work. And we have combined several similar results into one result as soon as possible. The content of result section has been revised in manuscript (The revised details can be found in Line 180-183, page 5Line 212, page 7).

Point 4: There are some grammatical problems, please fix them. "content" has been written two times in Fig. 2D.

Response 4: I'm sorry we don’t find the details in the manuscript before your reminder, and we have deleted repetitive “content” in Fig. 2D. (The revised details can be found in Line 194, page 6).

Point 5: Not sure about the number of pots, if one pot is considered as one replicate, you will have three pots for each treatment while you have mentioned 18, it is confusing. Did you have pseudoreplicate?

Response 5: I'm sorry that our expression was not clear enough in the manuscript. In fact, six treatments with 108 pots were employed in the experiment. There are 6 pots for one replicate, and three replicates for each treatment, so it is 18 pots for each treatment. (The revised details can be found in Line 119-121, page 3).

Point 6: In Methods section there is no specific explanation regarding Citrulline measurement. You need to mention which amino acids you measured.

Response 6: It is true that citrulline measurement was missed out in the manuscript, we have supplemented it in mothed section of revised manuscript (The revised details can be found in Line143-144, page 4). Also, amino acids of the manuscript were total free amino acids, which were important nitrogen metabolites, and we have added “total free” word in revised manuscript (The revised details can be found in Line 158, page 4).

Once again, thank you for your precious advice, I have found my shortcomings in my current work. I will follow your advice to improve my scientific research level in the future.

 With best wishes,

 Yours sincerely,

 Weijie Jiang

Institute of Vegetables and Flowers, Chinese Academy of Agricultural Sciences

12   Zhongguancun S. St., 100081,Beijing,China

Reviewer 2 Report

A huge amount of work, need to correct the presentation, see my notes.

Review

Agronomy 502804, Chao Ma et al.

 ·         Line 13, 18, 23, 28 ’-’ and ’+’ indexes are not in a good place, the correct form is NH4+-N instead of NH4+-N and along the full manuscript

·         Fig 1 no needs for this figure, too general, Line 66

·         space between the cited numbers sometimes have (Line 71, 74) sometimes no have (35, 36, 59, 69, 301) need to correct according to the requirements

·         Line 121: „three replicates, and there were 18 potsfor each treatment.” It is not clear, if we have AN(control 1), NN(control 2), UN, AN-NN, NN-UN, AN-UN these are 6 different ’treatment’, if we have 3 replicates it means 18 pots altogether, and not „for each treatment”

·         Line 124 ’estimation’ word is inaccurate, ’measurement’ is better

·         Line 125 and 129 spelling mistake: „weighed” instead of weighted

·         Line 175 Statistical analyses: more details needed, what kind of test was applied to show the significant differences?

·         Line Fig.2 The title of figure not ready, more details needed. n7: s.e.? or s.d.? abbreviations? What does the small letters mean? Need to write these details in the cases of all figures.

·         Line 230 SPAD-502 can measure ’only’ the relative chlorophyll contents, which is not equal with absolute chlorophyll content. Please insert into the relevant places the ’relative’ word to the chlorophyll content.

·         Line 234 No any sentences about the calculation of correlation, please insert the details among the statistical descriptions.

·         Line 240 The title of table, also too long, please no write abbreviations in the title. Insert the unit to the measured parameters.

Need to check the list of references: a lot of typing problem, mainly spaces…+ Latin name of species is always italic (Line 471, 522)

Author Response

Dear reviewer

   Firstly, I would like to express our sincere thanks to you for the constructive and positive comments.

·Point 1: Line 13, 18, 23, 28 ’-’ and ’+’ indexes are not in a good place, the correct form is NH4+-N instead of NH4+-N and along the full manuscript

Response 1: Thank you for your good suggestion. ’-’ and ’+’ indexes are not in a correct place, which is led by typeface of word. ’-’ and ’+’ indexes is correct in Times New Roman, while they seem not in a correct place in Palatino linotype, which latter is the typeface required by Agronomy. We have revised ’-’ and ’+’ word typeface of abstract section, but the typeface of other section have been not changed, because we afraid the change of typeface maybe affect paper typesetting (The revised details can be found in Line 13,18,23,29, page 1).

 Point2: Fig 1 no needs for this figure, too general, Line 66

Response 2: Thank you for your good suggestion. We have deleted “Fig. 1 Glutamate cycle pathway of nitrogen metabolism and changed subsequent figure No.

·Point 3: space between the cited numbers sometimes have (Line 71, 74) sometimes no have (35, 36, 59, 69, 301) need to correct according to the requirements

Response 3: Thank you for your reminder. We have removed the space between two cited numbers. (The revised details can be found in Line 70, 73, page2).

·Point 3: Line 121:three replicates, and there were 18 potsfor each treatment.” It is not clear, if we have AN(control 1), NN(control 2), UN, AN-NN, NN-UN, AN-UN these are 6 different ’treatment’, if we have 3 replicates it means 18 pots altogether, and not „for each treatment”

Response 4: I am sorry that our unclear expressions confuse you, and we have changed the expressions in manuscript. “Six treatments with 108 pots were employed in the experiment, and 18 pots for each treatment in three replicates, including six pots randomly in each replicate.” (The revised details can be found in Line 119-121, page 3).

· Point 5: Line 124 ’estimation’ word is inaccurate, ’measurement’ is better

Response 5: Thank you for your good suggestion. “Estimation’ has been instead of “measurement” in revised manuscript. (The revised details can be found in Line 124, page 3).

· Point 6: Line 125 and 129 spelling mistake: „weighed” instead of weighted

Response 6: I am sorry that we may have different understanding on words “weighed” and weighted, we found “weigh” was treated in verbs but weight in noun in general, so we think “weighed” was more appropriate in manuscript, which fruit of cucumber was weighed in Line 125 and 129, page 3.

· Point 7: Line 175 Statistical analyses: more details needed, what kind of test was applied to show the significant differences?

Response 7: Thank you for your good suggestions, we have replaced section of statistical analyses in revised manuscript.The analysis of the variance was conducted by ANOVA with  Tukey multiple range test at P < 0.05 in different N-forms . Analyses were conducted in three replicates per treatment. Pearson’s correlation was carried out using SPSS 22.0 software.” (The revised details can be found in Line 175-177, page 5).

· Point 8:  Line Fig.2 The title of figure not ready, more details needed. n7: s.e.? or s.d.? abbreviations? What does the small letters mean? Need to write these details in the cases of all figures.

Response 8: Thank you for your good suggestion. We have supplemented more details in the cases of all figures.Bars represent the standard error. Different letters above the bars indicate significant differences at P < 0.05.” (The revised details can be found in Line 198-200, Fig.1; Line 210-211, Fig.2; Line258-259, Fig.3; Line282-284, Fig.4)

· Point 9: Line 230 SPAD-502 can measure ’only’ the relative chlorophyll contents, which is not equal with absolute chlorophyll content. Please insert into the relevant places the ’relative’ word to the chlorophyll content.

Response 9: It is true that SPAD-502 can measure ’only’ the relative chlorophyll content, we added the ’relative’ word in revised manuscript (The revised details can be found in Line 144, page 4; Line 227,229,230, page 7; Line 401,403, page 14).

·Point 10: Line 234 No any sentences about the calculation of correlation, please insert the details among the statistical descriptions.

Response 10: Relationship between variables was tested by Pearson’s correlation. Pearson correlations were run between dry matter, biochemical substances, enzyme changes, nitrogen content and fruit yield, and a Pearson correlation coefficient (r) was derived. Correlation analysis was carried out using SPSS 22.0 software. (The revised details can be found in Line 176-177, page 5).

·   Point 11: Line 240 The title of table, also too long, please no write abbreviations in the title. Insert the unit to the measured parameters.

Response 11: Thank you for your good suggestion. The title of table 4 have been revised, and we added the unit to the parameters in the table. (The revised details can be found in Line 237, page 8).

  Point 12: Need to check the list of references: a lot of typing problem, mainly spaces…+ Latin name of species is always italic (Line 471, 522)

Response 12: Thank you for your good suggestion. We have checked and revised Latin name of references section (Line 479 and Line 521, page 15-16).

Thank you for your suggestions, which are instructive to our thesis writing and scientific research work, and we learn more knowledge from your valuable comments.

 With best wishes,

 Yours sincerely,

 Weijie Jiang

Institute of Vegetables and Flowers, Chinese Academy of Agricultural Sciences

12   Zhongguancun S. St., 100081,Beijing,China

Round  2

Reviewer 1 Report

Thanks for your response,

Please check lines 116-117. Both controls, 100% NH4-N and 100% NO3-N have been considered as No3-N treatment. Is it correct? It is not clear to me.

A grammatical mistake in line 217, Figures..... show (not shows).

Author Response

Dear reviewer   

      Firstly, I would like to express our sincere thanks to you for the constructive and positive comments.

     Point 1: Please check lines 116-117. Both controls, 100% NH4+-N and 100% NO3--N have been considered as No3-N treatment. Is it correct? It is not clear to me. 

    Response 1: I am sorry that our unclear expressions confuse you, and we have changed the expressions in manuscript (the revised details can be found in Line 116-117, page 3). Nitrate nitrogen (NO3--N) and ammonium nitrogen (NH4+-N) are two kinds of nitrogen used generally in vegetable planting. To find the effects with adding amide nitrogen to NO3--N and NH4+-N respectively, two comparisons were designed in this experiment: 100%NH4+-N was compared with 50%NH4+-N+50%CO(NH2)2, and 100%NO3--N was compared with 50%NH4+-N+50% CO(NH2)2, respectively.

     Point 2: A grammatical mistake in line 217, Figures..... show (not shows). 

     Response 2: Thank you for your good suggestion. “shows’ has been instead of “show” in revised manuscript. (The revised details can be found in Line 271, page 9). 

    Thank you for your suggestions, which are instructive to our thesis writing and scientific research work, and we learn more knowledge from your valuable comments. 

  With best wishes,

   Yours sincerely, 

   Weijie Jiang Institute of Vegetables and Flowers, Chinese Academy of Agricultural Sciences 12 Zhongguancun S. St., 100081,Beijing, China